# Suffering in silence: Accessing mental health care and repetitive transcranial magnetic stimulation (rTMS) for peripartum depression - A qualitative study

**Huda F. Al-Shamali**[1], **Rachael Dong**[1], **Margot Jackson**[2], **Lisa Burback**[1], **Gina Wong**[3], **Bo Cao**[1], **Xin-Min Li**[1], **Andrew J. Greenshaw**[1], **Yanbo Zhang**[1]*

**1** Department of Psychiatry, University of Alberta, Edmonton, Alberta, Canada, **2** Department of Nursing, MacEwan University, Edmonton, Alberta, Canada, **3** Faculty of Health Disciplines, Athabasca University, Athabasca, Alberta, Canada

* yanbo9@ualberta.ca

## Abstract

Peripartum depression (PPD) is a prevalent and serious mental health disorder that is often underdiagnosed and undertreated due to limited effective and safe treatment options. Repetitive transcranial magnetic stimulation (rTMS) has emerged as a non-invasive treatment for PPD, yet awareness among patients is low. This study aims to identify barriers and facilitators to accessing mental health treatment, particularly rTMS, for PPD. We conducted 36 interviews with individuals who experienced depressive symptoms during the peripartum period and health providers, followed by a descriptive interpretive thematic analysis. Key risk factors identified include personal (i.e., age), clinical (i.e., traumatic birth), situational (i.e., COVID-19, homelessness), and social (i.e., discrimination, domestic abuse). Five themes emerged regarding barriers and facilitators: 1) the need for mom-centered care, 2) systemic challenges, 3) the importance of mental health education, 4) stigma and custody concerns, and 5) challenges in accessing care. Eighty-three percent of participants were unaware of rTMS, but following a brief description, 75% were willing to receive or refer to rTMS if it was available to them. Addressing systemic and access-related concerns is crucial to ensuring patients with PPD have access to safe, effective, and accessible treatments.

## Introduction

The pregnancy and the postpartum period are pivotal phases in a parent's journey, often accompanied by an increased susceptibility to peripartum depression (PPD). PPD is diagnosed as a major depressive episode/disorder (MDE) with peripartum onset during pregnancy or within four weeks after childbirth according to the Diagnostic and Statistical Manual of Mental Disorders (DSM-5). MDE symptoms include persistent feelings of sadness, loss of interest or pleasure, disturbances in sleep and appetite, fatigue, feelings of worthlessness or excessive guilt, poor concentration, and thoughts of self-harm or suicide. Additionally, PPD may encompass unique manifestations specific to the perinatal period, such as disinterest in

**Data availability statement:** ***PA AT ACCEPT: Please follow up with authors for non-author contact available at accept*** Thank you for bringing this to our attention. We are currently in contact with our institution's REB to identify an appropriate individual or institution to act as the non-author contact. At present, as per our ethics application, all individuals with access to the data are listed as authors. We will ensure that a non-author contact is identified and approved by the ethics board before a final decision is made regarding this manuscript.

**Funding:** This study was supported by the University of Alberta Start-up Fund (RES0052505) awarded to Dr. Yanbo Zhang.

**Competing interests:** The authors have declared that no competing interests exist.

the infant, a sense of detachment from the baby, and persistent distress regarding maternal competence [1]. While up to 85% of new mothers experience the transient and self-limiting condition "baby blues," 29% of pregnant people, and 26% of people postpartum develop PPD, making it one of the most common medical complications associated with this period [2–4]. Certain predisposing factors further heighten the risk of PPD, including a personal history of mental disorder, exposure to stressful life events, poor social support, experience of abuse, marital discord, challenges related to childcare, exposure to second-hand smoke, sleep disturbance, and physical health comorbidities such as chronic physical health conditions, preeclampsia, and gestational diabetes mellitus [3]. Childhood trauma has also been linked to the development of PPD [5].

Despite its pervasive prevalence, PPD is frequently underdiagnosed and undertreated, with a staggering one in five women indicating they were never asked about their mental well-being during perinatal appointments [6]. Traditionally, PPD is treated using psychotherapy, pharmacotherapy, and electroconvulsive therapy (ECT) [7], but these modalities face challenges due to issues like patient reluctance, stigma, and limited effectiveness [8–16]. The suboptimal management of PPD is particularly concerning given its associated heightened risks of maternal suicide and infant mortality [17–19]. Considering the profound impact of PPD on both maternal and infant health, including maternal-infant bonding [18,20–25], there is a pressing need for improved screening, treatment options, and treatment delivery. Thus, understanding the determinants influencing access and receipt of care during this period is critical.

With growing interest in exploring alternative PPD treatment modalities that are safe, effective, and acceptable to patients, repetitive transcranial magnetic stimulation (rTMS) has emerged as a promising option over the past two decades [26,27]. rTMS is a non-invasive neuromodulation technique that uses magnetic field pulses to induce electrical current changes in target brain regions, most commonly the dorsolateral prefrontal cortex (DLPFC) [28]. There is evidence that rTMS exerts antidepressant effects by causing recurrent and consistent firing of coactive cortical neurons implicated in depression, thereby potentiating synaptic plasticity. rTMS may additionally have modulatory effects on meta-plasticity, the plasticity of synaptic plasticity [29]. In 2002 and 2008, rTMS was approved by Health Canada and the US Food and Drug Administration (FDA), respectively, as a treatment for treatment-resistant depression (TRD). In addition to TRD, rTMS has demonstrated effectiveness in treating bipolar depression [30], anxiety [31], OCD [32], and other psychiatric conditions. Compelling benefits of rTMS include its non-invasive nature and localized effect, which may be particularly desirable for pregnant or breastfeeding individuals concerned about medication side effects. Further, rTMS has a good safety profile and is generally well-tolerated, with most side effects being mild and transient. The most frequently reported side effects include temporary headaches following stimulation and scalp discomfort at the stimulation site [33]. Existing research on rTMS as a treatment for PPD demonstrates the promise of rTMS as a safe and effective treatment for depression with onset during pregnancy [34–41] and postpartum [42–46]. However, there are only two randomized controlled trials, one conducted during pregnancy [37] and one postpartum [45], both with small sample sizes, underscoring the need for more extensive trials to validate these findings. In Canada, rTMS is available as a treatment for depression and other various psychiatric conditions within both public health services and private clinics. However, its uptake among individuals with PPD unfortunately remains low. This underutilization raises the question of why more individuals with PPD aren't receiving rTMS and if this underutilization is a symptom of a greater problem that affects access to mental health treatment in general.

The objectives of the present study are twofold: 1) To understand the barriers and facilitators in accessing and receiving mental health treatment for PPD; 2) To explore the barriers

and facilitators in accessing and receiving rTMS for PPD specifically. To our knowledge, this is the first study that utilizes interviews to further our understanding of the possible barriers and facilitators to both peripartum depression treatment in general and rTMS utilization specifically. By interviewing health providers and persons with lived experience, this study offers a multi-dimensional discussion incorporating professional expertise and personal insights. With our findings, we aim to inform future initiatives that will contribute to developing more comprehensive and effective treatment strategies for PPD, thus improving mental health outcomes for mothers and their children. This study received ethical approval from the University of Alberta Research Ethics Office (Pro00114151).

## Methods

### Participant recruitment

Adult participants (18 years of age and older) in Canada who currently or previously experienced depressive symptoms during the peripartum period and health providers who offer mental health treatment for PPD were selected using a purposive sampling approach. Participants were recruited from December 2022 to August 2023 nationally through social media advertisements and a partnership with the Moods Disorders Society of Canada, a well-recognized organization with broad reach and credibility in mental health advocacy. Local recruitment initiatives complemented this national approach by targeting diverse and underserved populations through community organizations (i.e., Boyle Street, Terra Center), distribution of posters at health clinics and universities, and collaboration with local mental healthcare providers. There were no incentives associated with recruitment and participation.

### Ethics

Informed verbal consent was received from all participants. All participants received and read an information form explaining the study and their rights as participants prior to the interviews. At the start of each interview, important points from the information form were reiterated, and participants were asked the following two consent questions: 1) Did you read and understand the information form, and 2) Do you consent to take part in this interview? Participants needed to answer 'yes' to both questions to proceed with the interview. Their responses were witnessed by the interviewer and documented in the interview transcripts. This procedure was approved by the University of Alberta Research Ethics Office (Pro00114151).

### Data collection

An iterative process was used for interview design to capture a comprehensive understanding of participants' experiences with PPD, PPD treatment access, as well as perspectives and attitudes on rTMS. Initial iterations of the interview guide focused on rTMS only, but were insufficient to answer the research questions with adequate depth. In subsequent iterations, the flow of questioning was developed to naturally progress from participants' experiences with PPD, to treatments they accessed, to a discussion about rTMS. Interview questions were designed to be open-ended, allowing participants to direct conversation toward what they deemed most significant. This flexibility aligns with principles of qualitative research that prioritize participant perspectives and emphasize their lived experiences.

The final interview guide consisted of seven questions (Supporting Information File 1). The semi-structured interviews were conducted through Zoom by a single interviewer (PhD) until a point of data saturation was reached [47]. Demographic information, including age, ethnicity, gender, and the first three digits of their postal code, was also collected to identify

potential barriers to treatment access, such as distance from the center and transportation needs, as well as to assess how age, gender, or ethnicity may influence treatment choices and access. Thirty-six interviews were completed, with an average length of approximately 25 minutes ranging from 10 to 50 minutes. The interviews were video and audio recorded and automatically transcribed by Zoom. Transcripts were then verified for accuracy and anonymized by a researcher.

## Data analysis

The interview transcripts were thematically analyzed using the Braun and Clarke six-phase framework [48]. Two researchers were involved in thematic analysis; discussions took place throughout and disputes were resolved by reaching consensus. In the first phase, data familiarization, the researchers read through and verified the interview transcripts. Subsequently, the anonymized transcripts were uploaded to the NVivo software, where transcripts were read more deeply and inductive codes were generated that captured significant and recurrent concepts and ideas within the data. The NVivo software organized all similarly coded quotes, facilitating the process of organizing various codes into themes. Themes were then reviewed and supplemented with relevant excerpts from the transcripts.

## Results

### Demographic information

The mean age of the participants was 34.08 years [SD= 6.17] and ranged from 23 to 51 years old. All participants identified as women; most were Caucasian (75%) and lived in urban areas (92%; Table 1). There were 36 interviewees in total; eight of these were health providers, four of whom also had lived experience of depressive symptoms during the peripartum period. Of the 32 participants who experienced depressive symptoms during the peripartum period, five did not receive any treatment. Three others did not receive treatment during their first pregnancy despite experiencing depressive symptoms but were able to access care in subsequent pregnancies, either by being more proactive or through the support of a nurse who facilitated the process. Among the 27 participants who did receive treatment (medication: 20 [74%]; psychotherapy: 16 [59%]), many described the significant challenges they faced in accessing care. These challenges will be explored further in the 'Barriers and Facilitators in Navigating Peripartum Mental Health Care' section. Additionally, treatment was often delayed until after delivery (n = 15; 56%), and for those already on medication prior to pregnancy, treatment was sometimes halted until postpartum.

### Symptoms, intrusive thoughts, and suicidality

A wide array of symptoms were reported by participants showcasing the various emotional and physical reactions invoked by PPD. Feelings ranged from sadness and an inability to control their emotions to a profound sense of emptiness and numbness. Participants also reported feeling overwhelming heaviness and exhaustion and a deep-seated sense of vulnerability. Guilt, anxiety, hopelessness, embarrassment, and loneliness are just a few of the many other symptoms brought up during the interviews. One participant even mentioned that they no longer plan on having any more children due to the severity of the depressive symptoms they experienced with their first child. Others mentioned feelings of failure and feeling like a "bad mom" due to their PPD, which resulted in fear of admitting to others that they were depressed.

Intrusive thoughts were also commonly experienced, with 14 participants discussing occurrences that ranged from thoughts related to self-harm or harming their child to suicidal

**Table 1.  Participant characteristics.**

| Province/Territory | | AB | BC | SK | MB | ON | QC | NB | YT | N/R | Total |
|---|---|---|---|---|---|---|---|---|---|---|---|
| All Interviewees | Total (n): | 21 | 2 | 4 | 1 | 4 | 1 | 1 | 1 | 1 | 36 |
| Age (mean [SD]) | | 36 [6] | 37 [6] | 29 [4] | 35 | 30 [5] | 30 | 34 | 41 | 27 | 34 [6] |
| Ethnicity | | | | | | | | | | | |
| | Caucasian | 16 | 2 | 4 | 1 | 1 | 1 | 1 | – | 1 | 27 |
| | Asian/Pacific Islander | 1 | – | – | – | 1 | – | – | – | – | 2 |
| | First Nations/Metis/Inuit | 2 | – | – | – | – | – | – | – | – | 2 |
| | Latina/Hispanic | 1 | – | – | – | 1 | – | – | – | – | 2 |
| | Mixed (First Nations & Caucasian) | 1 | – | – | – | 1 | – | – | – | – | 2 |
| | Prefer not to say | – | – | – | – | – | – | – | 1 | – | 1 |
| Community | | | | | | | | | | | |
| | Urban | 20 | 2 | 4 | 1 | 3 | 1 | 1 | 1 | – | 33 |
| | Rural | 1 | – | – | – | 1 | – | – | – | – | 2 |
| | N/R | – | – | – | – | – | – | – | – | 1 | 1 |
| Treatments Received | | | | | | | | | | | |
| | Medication | 10 | 2 | 3 | 1 | 2 | 1 | 1 | – | – | 20 |
| | Psychotherapy | 10 | – | 2 | 1 | 3 | – | – | – | – | 16 |
| | None | 2 | – | 1 | – | – | – | – | 1 | 1 | 5 |
| Health Providers | Total (n): | 4 | 1 | 2 | 1 | – | – | – | – | – | 8 |
| Professions | | | | | | | | | | | |
| | Social Worker | 1 | – | 2 | – | – | – | – | – | – | 3 |
| | Nurse Practitioner | 1 | – | – | 1 | – | – | – | – | – | 2 |
| | Therapist | 1 | – | – | – | – | – | – | – | – | 1 |
| | Registered Nurse | – | 1 | – | – | – | – | – | – | – | 1 |
| | Professor | 1 | – | – | – | – | – | – | – | – | 1 |

Abbreviations: AB, Alberta; BC, British Columbia; MB, Manitoba; n, sample size; NB, New Brunswick; N/R, Not reported; ON, Ontario; QC, Quebec; SK, Saskatchewan; YT, Yukon.

ideation (Table 2). One participant even shared that they attempted suicide after ceasing their medication treatment, believing they had improved and without receiving any subsequent follow-up. Upon discontinuation of the medication, their symptoms quickly returned, leading to the suicide attempt.

## Framing the issue of PPD through lived experience

To properly understand the barriers and facilitators to PPD treatment, it is essential first to be aware of the situations that women experiencing PPD face, which may have contributed to the onset of their depression and continue to affect them. The diverse experiences shared by participants in the interviews reveal a complex interplay of personal, clinical, situational, and social factors in the emergence of PPD, complicating the process of accessing and receiving care (Table 2).

At the personal level, the mother's age, whether an older (>35 years) or young new parent (<20 years), was mentioned as a key factor that complicated their ability to transition into parenthood. An older mother shared that this difficulty stemmed from never having experienced such dependence from another person. For young parents, the difficulty often stemmed from a lack of support. A therapist specialized in working with young parents highlighted that they often lack the support older parents receive and instead encounter judgmental attitudes, such as being told, "You made your bed, now you have to deal with being a parent."

**Table 2. Lived experiences of PPD - supporting quotations.**

| | Supporting Quotations |
|---|---|
| **Symptoms, intrusive thoughts, and suicidality** | |
| **Intrusive thoughts** | "I definitely had, like, "Oh my God it would be so great if I could just get like seriously injured", like by, like hit by a car so that I could have a break and like be in the hospital and be immobilized, and not have to take care of anything." (Participant 17)<br><br>"I was having thoughts that I was gonna get up and put my baby in a snowbank, in the middle of the night, sleeping, and I would never do that. But it's not that those thoughts don't happen to everybody, but once it was there I couldn't get rid of it…Unfortunately, the first time I had postpartum, was the first time I ever contemplated suicide my entire life, and I am not, I am terrified of death… I said if it ever came down to it, I would just go instead of [my baby]." (Participant 10) |
| **Suicidality** | "The feeling of, you know, like not being enough, or not being what my daughter needed me to be, was, was terrible…I just had very intrusive thoughts of like, if we, if the three of us just die, you know, like that would solve everything because then we don't have to experience this. We don't have to go through it. It's better if we just die. But like, I didn't want to die alone. I just wanted the three of us to die at the same time, so we can, I don't know reunite in heaven, or whatever, you know, like, I didn't even know what my expectation was, but I just wanted to be together, but in a state of peace, and I thought that dying was the solution. I never came up with a plan to kill the three of us, or anything like that, or myself, or harm myself. Umm, but I, I was hurting myself in the sense of like, you know, when I made a mistake, when I spilled some milk, or when I didn't do something, that was like exactly right, I would like hit my head so hard, and be like, why are you so stupid, and treated myself like that because I thought it was all my fault, right, that this was happening to us." (Participant 13) |
| **Framing the issue of PPD through lived experience** | |
| **Personal Factors** | "I found the transition into motherhood very difficult, because until that point, I had not had anybody rely on me in that kind of nature… I really struggled with it… and then, of course, I struggled with struggling with that." (Participant 1)<br><br>"Older parents receive more help from other people. And there's this expectation that parenting is difficult… that's not something that's common in the adolescent led parenting world. I think, umm, many of my young parents receive messages from others like "well, you made your bed, so now you have to deal with being a parent"…and really penalizing our young people for needing what everybody [laugh] needs." (Participant 32) |
| **Clinical Factors** | "I had a really traumatic birth.. I hemorrhaged after umm, my C-section and, umm, I was rushed to the ICU… I personally attribute my, umm, postpartum depression to, to that…as soon as I got discharged, umm, I noticed, umm, my mood was pretty bad, umm, and it started to get worse… It wasn't normal." (Participant 20) |
| **Situational Factors** | "Essentially all of our clients, umm, come in with like some level of depression, mental health issues, etc. But like, just given their vulnerability, their circumstances, usually they're homeless, like those kind of things go hand in hand, right.…People need stability in order to be able to take care of themselves. And if they're living on the streets, they have no sense of security, no sense of safety, no stability. So getting clients housed, can mean the difference of like, somebody going to detox and treatment and getting off substances and being able to parent baby versus not, right?... And then the next biggest factor is having good supports in place…When we have clients that are [in homeless shelters], they're still literally in crisis... It's very invasive. They really need their own housing." (Participant 34) |
| **Social Factors** | "After the gender reveal, uh there was great disappointment, gender disappointment, because it was a baby girl instead of a baby boy…I wasn't disappointed, but my partner was, and uhh, he… umm, made me guilty for not, umm, producing the gender he wanted… the abuse started since the week 22, pretty much. So I was trapped in a situation where I was super pregnant and could not leave, and that led to incredible, uhh, feelings of anxiety and depression… After that, there was extremely traumatic birthing experience, uh five days in labor…The opiates, it just makes you limp. And first few months of, umm, my baby's life outside my womb, uhh, I don't even remember that much, because I do remember verbal abuse, I remember being, uhh, made to do things I didn't want to do… M- my, uh, biggest struggle was, uhh, domestic abuse I experienced, and just to struggle through it. Uhh, pretty much locking myself in the room to cry, uh yeah, and uhh, without any way of reaching out for help. I didn't even know the resources, because, uhh, it was my beginning of the third year of, in Canada, so I wasn't aware of what- where can I go… I did not feel like I can reach out to anyone. It's the way this thing works. They isolate you. They just keep you under their control, and leave you crazy, right?" (Participant 19) |

From a clinical standpoint, experiencing hormonal fluctuations as well as a history of psychiatric conditions were common in our sample, with several participants previously suffering from depression, anxiety, an eating disorder, substance use disorder, ADHD, or PTSD following sexual assault. Another common occurrence was traumatic/complicated deliveries or their child being born with a birth defect.

Situational factors that exacerbated participants' vulnerability to depression were the fear and isolation caused by the COVID-19 pandemic, unexpected pregnancy, and homelessness. Homelessness, in particular, creates instability and a lack of privacy, which significantly increases the risk of depression. A nurse who works primarily with unhoused pregnant and

postpartum women highlighted how these conditions contribute to feelings of vulnerability and lack of security, making depression more likely.

Finally, social factors, such as having no support, experiencing discrimination, and being in an abusive relationship, all played critical roles in participants' experiences with PPD.

## Barriers and facilitators in navigating peripartum mental health care

When discussing participants' ability to access and receive treatment, five themes highlighted the challenges and potential avenues for improvement. The themes include (1) a need for mom-centered care, (2) systemic challenges in peripartum mental health, (3) the importance of mental health education, (4) stigma and custody concerns, and (5) challenges in accessing care (Table 3).

**A need for mom-centered care.** Some participants' experiences attending prenatal and postpartum appointments were described as "dehumanizing" and "devaluing". These individuals often felt that the care provided was disproportionately focused on the fetus or baby, leading to a situation where their health, both mental and physical, was made secondary. This fetus/baby-centered approach made participants feel overlooked and took away their feelings of autonomy during a period that requires extensive support. Other participants felt seen during pregnancy but felt no longer prioritized as soon as the baby was born. A positive feature of the current postpartum care process that was mentioned was receiving care from midwives and having a nurse conduct home visits. Having these support persons was seen as beneficial and one-way participants felt cared for and supported.

**Systemic challenges in peripartum mental health.** Participants identified three systemic challenges that affect a person's ability to receive the mental health care they need during the peripartum period. The first barrier discussed was encountering dismissive and unsupportive health providers who completely overlooked or minimized the depressive symptoms to "normal" symptoms that are just part of pregnancy and motherhood and that they just needed to be endured. Due to this, some participants felt less empowered to share their concerns with their doctors since they did not feel seen or cared for. Additionally, the dismissive attitudes of some healthcare providers towards mental health issues contribute to a breakdown in trust. This skepticism results in some participants choosing not to disclose their depressive symptoms, anticipating a lack of support or understanding from their healthcare providers.

The second aspect of the healthcare system that participants would like to see improvement in is the mental health screening process during the peripartum period. Some participants had a positive experience and received resources or treatment due to the mental health screening completed by a nurse. However, this is not the case for everyone. Others were either not screened at all, dismissed and told, "if you're gonna kill yourself, call someone", or screened and offered no further support.

The third point made was a need for more proactive and preventative approaches to mental health care that are patient-centered. One suggestion to make peripartum mental health care more proactive involves creating treatment plans before the baby is born and informing patients about the resources available to them. This improves individuals' awareness of their options for support and treatment ahead of experiencing depressive symptoms, thereby facilitating earlier intervention and preparedness.

**The importance of mental health education.** Across the interviews, there was a pronounced need for more education regarding the distinction between normal versus abnormal symptoms during pregnancy and postpartum. Additionally, participants expressed a desire for more information on available services and the process for accessing them, as well as guidance on how to support themselves and their partners.

**Table 3. PPD lived experience - supporting quotations.**

| Theme | Supporting Quotations |
|---|---|
| **1. A need for mom-centered care** | "It's a very, like, powerless process in a lot of ways…You feel like you can't make any decisions, like, the getting, like, weighed every week and getting my, like, stomach measured every week. And I was like, "is this actually necessary?"… I felt like livestock… It was like a very dehumanizing experience. Umm where I felt like… the baby was the patient, and I was like the vessel… Umm, I think having more access to, like a patient-centered approach… As a pregnant person, I had a lot more insight about what was going on in my body then a measurement was gonna say right… It all felt like just super super procedural, and like quite rigid and quite cold." (Participant 4)<br>"When you're pregnant, you're treated so well. Because, you know, doctors and, and health care professionals really care about having a healthy baby, and Mom needs to be healthy for that to happen. But then, as soon as that baby comes out, you're like, it's not important anymore… people want to keep babies alive and healthy, but it, you know, until it's too late, mom doesn't really get that kind of support." (Participant 31)<br>"I don't even remember what it's called, but it's, it's that program where, like the nurses follow you, and they visit your home after you've had a baby. That one seems to be a big hit. People really like that the nurses do seem extremely empathetic and supportive and, and more concerned about mom and baby as opposed to just baby." (Participant 31) |
| **2. Systemic challenges in peripartum mental health** | **2.1 Dismissive/unsupportive health providers**<br>"Getting on medication was very difficult. I phoned my family doctor, and his suggestion was, umm, that I'm probably too fat, so that I need to do yoga, and that will make me feel better. And I was like, umm, you know, I'm suicidal. So like, [laugh] we're not talking about yoga anymore. Umm, and then I very, very like directly said, I want you to put me on an antidepressant." (Participant 36)<br>"I had a family doctor, but it wasn't a person who I felt like I could share these kinds of things with him… I felt like he was like a business model... They are like typing in their little computer all the time, and then they prescribe you something, and it's like, okay, go away, you know. Like, you don't feel that they are invested in your health, or that they care about you…I didn't even know his name, you know. Umm, I doubt that he knew mine." (Participant 13)<br>"The other big barrier too is… so I complain about how tired I am and how much, you know, I'm, I'm having these secret thoughts of wanting to hurt my baby or running away, or of giving my child up, or kind of any of these concerns, or not being a good enough parent to them, umm, that they don't need me, that they don't want me… "what are you gonna do about it?" So, there's this sense of like, if I, if I tell you then now, not only are you gonna think these bad things about me, the stigma, but you can't do anything anyway. So, umm, asking for help will just be even more futile than if I kind of just suck it up and deal with it on my own." (Participant 32)<br><br>**2.2 Mental health screening process**<br>"And we do the little screening for postpartum, she said, "yeah, I feel all those things. What are you gonna do? Are you gonna send someone to my house? Do you have like support and help? Are you gonna send a wet nurse, because I'm not sleeping, because I can't, because my child nurses constantly and that's part of why I'm exhausted like, how are you going to fix this problem?" So it's not just enough to check the checklist. But what are you gonna do about it, so that I just don't feel even more helpless when I leave that now you've, you know, said that I'm at moderate risk or at, you know…serious, severe levels, umm, of concern, but then what? I just get to walk out of here and with your assessment, and [laugh] just know that, that's what I'm dealing with." (Participant 32)<br><br>**2.3 Proactive/preventative care needed**<br>"Umm, I think, I think the resources need to come to the pregnant people, umm, or the, like people who just had babies. Umm, they need to be specific to the demographic or the need, right… Also, like a really big part of it, is that like, screening for domestic violence needs to be clearer, right? Because, like, how many of these women, umm, are, are accessing these services while there's issues going on at home, which are exasperating them. So, it's really, I think, it's not up to the people suffering, to, in like, the haziness of, of pregnancy or, or, or newborns, to, try and find those resources, because they're not going to do it." (Participant 31)<br>"If there's any way to have, like some sort of support beforehand, to prepare the person mentally, whether that be like, umm, I don't know, like even therapy sessions, or something like that, where you get to talk to a professional before you have a baby [laugh]. Umm, so that you know, as soon as you see there's issues, you can go to them, and you already have that system in place." (Participant 20) |
| **3. The importance of mental health education** | "He is supporting a person through depression postpartum and-and current, umm pregnancy depression. That needs to be addressed for men, because I feel like, you know, I would like to know the statistic of how many people divorce after 2 years, after having the baby. Because it's so hard, and they don't know how to, they don't have the tools to deal with that." (Participant 10)<br>"I know a lot of the girls, especially with, like, interactions with the medical system, umm, many of the physicians and nurses do way too much medical talk. They talk over them. It's not plain language and the clients have no idea what they're being asked about. They get really scared, they close off, they feel like they don't have any control or autonomy over their decisions because they just, they wanna understand what's going on, but they just don't because they're not being talked to in a way that they can really like understand what's going on." (Participant 34)<br>"A lot of times what they just do is they give out a pamphlet…but at the time when I came home with my son, I had so many pamphlets, and also my brain was mush and I couldn't, nothing was sticking in my head. I was very overwhelmed, and I didn't feel like I could read or take things in… and I needed somebody to sit and, and to not just throw information at me, but to talk with me, and to ensure that I actually heard or understood the information." (Participant 21) |

*(Continued)*

**Table 3.** (Continued)

| Theme | Supporting Quotations |
|---|---|
| **4. Stigma and custody concerns** | "I don't know if this one is something that could really be done, but there's still that high stigma soc- umm, socially on, umm, on mental health issues, and that you're weak or something is wrong if you have an issue, right? So it's just a lot less likely that you're going to reach out for help. Because even as a new mom, it feels like you fail if you have something… an issue." (Participant 23)<br>"I definitely want this on the record. We gotta stop taking away people's children… I have a friend for my last due date group…she umm, had to admit herself, but, like she doesn't get to see her child now, and it just, it almost seems counterintuitive, like it almost just seems like, yes, she has to get her health in order, like, but how do you do that when you can't be a parent, and you just had a baby. And I also think back to being like, okay, I can't tell anybody, I'm feeling this way because somebody is going to take my kid." (Participant 8)<br>"They are scared about the stigma. They have an abusive partner who tells them that if they seek help that will look bad and they'll get the kids taken away. Umm, that's another one, too, would be like, especially in our indigenous population with, umm, there's been, obviously high-, like higher rates of like apprehension over the years, so…they would be especially worried of, umm, the risk of apprehension too…So they just don't seek help at all, because they think it's not an option." (Participant 35)<br>"Women, we are umm, protective by design, of, especially of our kids. And I feel like, when somebody hands you a number [laugh], there's a part of you that's afraid they are like, are they assessing me? Are they gonna take away my baby [laugh]. Like this is the crazy stuff that goes through your head when you're in those types of situations." (Participant 10)<br>"I attended a virtual support group, and I had been nervous about going to it because I, it was with the health, the health authority, and I wanted to like get a job with the health authority, so I was concerned about, like, like potential future coworkers being the ones providing the service. Umm, I also felt nervous about, like who else would be in there? Umm, like in previously, umm, when I first knew I was pregnant, I was worked, I was in Manitoba, and I was working with pregnant women, and I was concerned about going to like prenatal classes that were free, because I was concerned that my clients would be there." (Participant 21) |
| **5. Challenges in accessing care** | **5.1 Depressive Symptoms** |
| | "Just getting out of the house was so hard. Umm, and of course, just with my mindset too, just everything hurt, and, and it was just hard and just felt heavy. Like, I was, I felt very fortunate if I could get to an appointment, and so I, I just wasn't able to keep it up, like it was just, it was just too hard for me to get there… it would have been so nice to just have something set up prior to the birth. Umm, so that I didn't have to do any of that thinking or searching or finding, 'cause I, I just wasn't really in a mindset to even know what I needed, right?" (Participant 1) |
| | **5.2 Physical accessibility** |
| | "I never got any sort of a psychiatric referral. Umm I live in the North, umm so our psychiatric services are really limited here anyways, umm like my referral probably wouldn't have, I wouldn't have gotten an appointment til after I had my baby, anyway." (Participant 4) |
| | **5.3 Limited availability of doctors** |
| | "And then I guess, yeah, for people who don't have access to a family doctor, like that would be a huge thing. That's probably a big miss for a lot of people, is that in this day and age where most people don't have a family doctor, umm, yeah, like they, what would happen to them if they were depressed? And they, they have no one to ref- like the, they screened high, but there's no one to refer them to." (Participant 27) |

Health providers' knowledge of PPD and the availability and accessibility of treatments were considered crucial as they are often the patients' primary source of knowledge and guidance when accessing treatments. Furthermore, having health providers communicate with their patients in a way that is easily understandable and avoids the use of complicated medical jargon was considered important.

Participants suggested that receiving resource lists or having access to an "organized database" with all the services they may need, mental health-related or otherwise, would be a great step towards improving their awareness of what they are experiencing and what they can do about it. However, this was not considered sufficient by some participants who prefer information to be presented through a conversation with a health provider who can check their understanding.

**Stigma and custody concerns.** Stigma surrounding mental health was discussed as a barrier that discouraged individuals from seeking support or treatment during the peripartum period. A grave concern brought up in the interviews related to the stigmatization of mental health was a fear that they would lose custody of their child or children if they mentioned that they were experiencing depressive symptoms. One participant reported that this fear was rooted in the fact that this had happened to her friend. This fear also translated into

apprehension towards the intention behind postpartum mental health screenings and whether screening as a high risk for depression would get them "into trouble".

For participants who were both health providers and had experienced depressive symptoms during the peripartum period, an additional layer of complexity arose regarding seeking treatment. Given their professional roles, where they frequently refer patients to mental health services or where their potential treatment providers could be their colleagues, there was a notable hesitancy to access services.

**Challenges in accessing care.** Accessing mental health care was considered exceptionally challenging due to three primary barriers. The first barrier was the depressive symptoms themselves, which made taking any steps towards receiving treatment overwhelming and seemingly impossible, as simply existing already seemed like a lot to handle. The second barrier was concerning the process of physically accessing a treatment. Factors that limited physical accessibility were the individual's location (i.e., lives in a rural community), long waitlists, need for child care, availability of transportation, the COVID-19 pandemic, and the cost of the treatment. The third barrier was the limited availability of doctors, particularly in rural areas. This limited the continuity of care and prevented the individual from building a trusting relationship with a consistent doctor.

## Repetitive transcranial magnetic stimulation (rTMS)

Thirty participants had never heard of rTMS before this study, and for those who had heard of rTMS, the majority did not know many or any details about this treatment modality. The relative attitude towards rTMS, following a brief description provided by the interviewer, was positive, with 27 of the participants indicating that they would be interested in receiving or referring others to receive rTMS during the peripartum period. Twelve participants indicated that they are more willing to receive rTMS than medication, and only one participant stated that they would rather try other treatments first. One of the participants who preferred rTMS over medication said:

> Yeah. I would have done that when I was pregnant, 100%. Yeah, I would have, to get rid of that feeling, I would have done anything. And, like I said, I just I couldn't, I couldn't do the medication for personal reasons. I just wasn't comfortable with [my baby] having medication in her, umm, but that I would have done. I would have absolutely done that. (Participant 10)

Additionally, two participants stated that they would be willing to receive rTMS during pregnancy but would be more hesitant to receive rTMS during postpartum due to not having the time to access it then. Conversely, two participants preferred to receive rTMS postpartum due to fear of its potential impact on their developing fetus. Overall, there was a desire to learn more about rTMS, its mechanism, safety, and effectiveness.

Two participants stated that they are not willing to receive rTMS; one participant due to the extensive time commitment, and the other due to a preference for more typical treatment modalities:

> Umm, personally, I'm old school, I would take the old route, like, you know, counselling, maybe medication or something, eventually get back off the medication... Umm, so if a person is open to it, sure, but I wouldn't be. (Participant 33)

The remaining seven participants expressed hesitancy to receive or refer rTMS due to the following reasons: it seems "odd", may be interpreted as invasive, need childcare, its availability in their community, and a need to know more about it and the research. Additionally,

a nurse who primarily works with unhoused pregnant and postpartum women stated that individuals first need a certain level of stability prior to being able to access rTMS consistently:

*The client who's here today…she's become very stable. So she would be a good candidate for sure… The majority though are not clients that would be suitable because they don't, they are not actively engaging enough that, that would be something they could follow through on… Whether it's cause they're not housed and can't be contacted, umm, whether it's substance use getting in the way, variety of reasons. (*Participant 34)

Two primary concerns regarding receiving rTMS during the peripartum period were discussed during the interviews.

**Accessibility concerns.**  The first concern was whether they would have access to this treatment, particularly if they lived in northern territories in Canada or rural communities. Some participants said they would be willing to receive rTMS if they didn't need to drive many hours or move temporarily to another area to receive it. Cost was also a major consideration, with multiple participants asking whether this treatment was covered by public healthcare. Also, the referral process and whether they could even access a doctor, not to mention one who knew about rTMS and could refer them, was a concern. Access to childcare and whether they could bring and breastfeed their baby during the appointment were also important components of accessibility that were discussed.

**Time commitment concerns.**  The time commitment of rTMS was mentioned as a possible barrier, particularly for persons with limited family support. Conversely, some participants viewed the time commitment as a positive as it would allow them to get out of the house and have time for themselves:

*Just the act of having to go to these appointments and having something on the calendar would have been really good. And, umm, I also think [laugh] that, umm, like having to have somebody baby sit your child… I think it would also have been good, you know, to kind of have forced me to take that time. (*Participant 21)

## Discussion

The interviews highlighted multiple personal, clinical, situational, and social risk factors that can increase a person's vulnerability to developing PPD, alongside a discussion on various symptoms, including intrusive thoughts and suicidal ideation. Several key themes arose, reflecting the challenges and potential improvements within the current framework for accessing peripartum mental health care: a need for mom-centered care, the importance of more responsive healthcare providers, enhanced mental health screening processes, improved mental health literacy, a need for efforts to de-stigmatization mental health, limiting child apprehension, and improving access to care. Additionally, most participants were unaware of the existence of rTMS, but responded positively after receiving a brief description of this non-invasive treatment modality. Some accessibility and time commitment concerns were mentioned as possible barriers to receiving this treatment. Despite these potential barriers, 75% of the participants were still interested in receiving or referring patients to rTMS if available.

The peripartum period is a particularly vulnerable state, as this is a significant life transition with concomitant hormonal and neurological shifts that increase the risk of developing depression and anxiety [49]. Moreover, as our interviews confirm, intrinsic factors such as age, and a history of psychiatric conditions, particularly depression and anxiety, significantly

contribute to the development of PPD [49,50]. Extrinsic factors, including the stress and isolation exacerbated by the COVID-19 pandemic, unexpected pregnancies, homelessness, a lack of support, experiences of discrimination, and being in abusive relationships, further compound this vulnerability [3,50,51].

This transitional period requires a greater emphasis on the mother's mental health rather than just the developing fetus or baby, as extensively repeated across the interviews. This begins with improved mental health screening and allowing space for patient-led conversations surrounding their pregnancy and postpartum experiences [52]. The importance of screening is further highlighted by the recommendation made by the American College of Obstetricians and Gynecologists (ACOG) that recommends the screening of depression and anxiety at least once during the peripartum period using validated screening tools [53]. The Canadian Task Force on Preventative Health Care takes a different approach by recommending against the reliance on depression screening questionnaires with cut-off scores as a means of assessing all individuals during the peripartum period. Rather, there is an assumption that health providers will allow space during appointments for mental health-related conversations to arise and remain vigilant to detect well-being-related concerns [54]. This recommendation responds to the need mentioned by participants to allow space for empathetic conversations focused on their mental health. It addresses their concerns regarding poor screening methods that do not always capture their experiences. However, is the assumption that space is currently available for these conversations during appointments true? Participants' accounts of encounters with dismissive health providers, feelings of being only a "vessel", and prevailing mental health stigmas suggest that we may still have a long way to go. This finding is supported by a 2018 qualitative study that assessed factors that impacted access to mental health care in individuals receiving midwifery care in Ottawa, Ontario. Participants in that study also reported feeling that perinatal healthcare is baby-centered and that there is a lack of services that focus on the mother [55]. Such findings indicate a persistent gap in the perinatal healthcare system's ability to integrate and prioritize maternal mental health fully.

Limited awareness of the existence and availability of rTMS is a major barrier to receiving this treatment. The interviews showed that 83% of the participants had never heard of rTMS before this study. Yet, following the interviewer's brief oral description of this treatment, 75% of the participants were willing to receive or refer others to rTMS if it was available to them. A previous study surveyed 51 pregnant women and found that 0% of the participants were willing to receive rTMS. However, after a brief informative video on rTMS, 15.7% indicated they would receive rTMS [56]. This shows that providing patients with information on rTMS can improve their acceptance. Especially when the information is provided orally, which allows space for the individual to ask clarifying questions.

In our study, particularly in severe cases of depression, the willingness to try rTMS stems from an interest in its non-invasive nature and its potential to rid them of the depressive symptoms that significantly impact their lives. Despite the accessibility challenges associated with rTMS, such as the need for frequent sessions, geographical limitations to treatment centers, and the need for childcare, the awareness of rTMS as an available option is valued by those struggling with PPD. While rTMS may not be suitable for everyone due to these barriers, it presents a viable alternative for individuals with treatment-non-responsive cases of depression or for those who are reluctant or unable to pursue conventional treatments due to concerns about the potential effects on their child. This highlights the importance of expanding the mental health treatment toolkit to include innovative and patient-centered options like rTMS, ensuring individuals with PPD have access to a range of therapeutic choices tailored to their specific needs and circumstances.

## Policy recommendations

To address the identified gaps and barriers in peripartum mental health care, several policy recommendations are proposed to ensure improved outcomes for individuals experiencing PPD. First, implementing mandatory, standardized mental health screenings throughout the peripartum period, with validated tools and provider training, can help identify those in need of support early. Screenings should include space for open-ended, patient-led conversations to capture nuanced experiences that traditional questionnaires may overlook. Furthermore, the screening process must consistently be followed by active facilitation of treatment or referral to supportive resources by healthcare providers, lifting the burden of navigating care from patients and ensuring they receive timely and appropriate support.

Second, expanding education and training programs for healthcare providers can enhance their ability to engage empathetically and address mental health needs beyond a baby-centered framework. This includes mandatory continuing education focused on maternal mental health literacy and de-stigmatization. Educated and empathetic healthcare providers can, in turn, better educate and empower patients to understand their mental health, navigate available resources, and advocate for their needs. This holistic support can foster a sense of trust and collaboration, reducing feelings of isolation and helplessness among mothers.

Third, increasing funding for peripartum mental health services and innovative treatments like rTMS is essential. This includes subsidies or insurance coverage for rTMS, and other treatments (i.e., psychotherapy), to improve affordability and investments in expanding treatment availability in underserved and rural areas. Additionally, public awareness campaigns are needed to improve understanding of rTMS and other maternal mental health resources, reducing stigma and increasing informed decision-making among patients.

Finally, policies that address broader social determinants of health, such as housing instability, access to childcare, and support for survivors of domestic violence, are critical to mitigating extrinsic risk factors. Ensuring interdisciplinary coordination among healthcare providers, social workers, and community organizations can provide a more holistic and supportive care network for vulnerable populations. By integrating these policy recommendations, maternal mental health care can become more comprehensive, equitable, and patient-centered, ensuring better outcomes for mothers and their families during the peripartum period.

## Limitations

This qualitative study has limitations. Due to the voluntary nature of the sampling strategy, there is potential for self-selection bias, whereby participants interested in mental health or research participation, or those of a certain socioeconomic status or with lower caregiving burden, may be more likely to sign up for an interview. Concurrently, because of reliance on social media advertising for the majority of participant recruitment, there may be an under-representation of individuals without internet access. To mitigate these issues, targeted local recruitment efforts were made, including an in-person visit to a community center that supports street-involved pregnant and postpartum women; expansion beyond local efforts could further enhance the sample's inclusivity and relevance. Additionally, all interested participants were extensively accommodated during interview scheduling with the flexibility to reschedule their interview dates and times as many times as necessary to ensure that scheduling did not pose a barrier to participation. Another limitation is the recruitment of only participants who live in Canada, possibly limiting the generalizability of the findings to countries with different healthcare systems and maternal mental health processes. Further, there was no longitudinal

follow-up on participants' subsequent treatment decisions, limiting understanding of long-term acceptance and uptake of rTMS.

## Conclusion

This article presents various personal, clinical, situational, and social risk factors as well as offers a glimpse into the various symptoms experienced during PPD, including intrusive thoughts and suicidal ideation. Access to effective mental health care during the peripartum period is limited by the insufficient availability of care that prioritizes the mother's well-being, poor screening practices, limited mental health literacy, stigma and fear of child apprehension, as well as limited accessibility to mental health treatment. rTMS as a treatment for PPD was met with a positive attitude from a majority of the participants, most of whom never heard of rTMS before this study. The accessibility and time commitment related concerns were mentioned as possible barriers to receiving rTMS. Regardless, most participants were interested in learning more about rTMS and potentially receiving or referring others to this treatment. Urgent measures are required to expand the capacity of perinatal appointments to address mental health-related concerns effectively and increase the knowledge of health providers and patients on available treatments and how to access them.

## Supporting information

**S1 File. Interview questions.**
(DOCX)

## Author contributions

**Conceptualization:** Huda F. Al-Shamali, Margot Jackson, Lisa Burback, Gina Wong, Bo Cao, Andrew J. Greenshaw, Yanbo Zhang.

**Data curation:** Huda F. Al-Shamali.

**Formal analysis:** Huda F. Al-Shamali, Rachael Dong.

**Funding acquisition:** Xin-Min Li, Yanbo Zhang.

**Investigation:** Huda F. Al-Shamali.

**Methodology:** Huda F. Al-Shamali, Margot Jackson.

**Resources:** Margot Jackson, Xin-Min Li, Yanbo Zhang.

**Supervision:** Huda F. Al-Shamali, Margot Jackson, Andrew J. Greenshaw, Yanbo Zhang.

**Validation:** Huda F. Al-Shamali, Rachael Dong.

**Visualization:** Huda F. Al-Shamali.

**Writing – original draft:** Huda F. Al-Shamali, Rachael Dong.

**Writing – review & editing:** Huda F. Al-Shamali, Rachael Dong, Margot Jackson, Lisa Burback, Gina Wong, Bo Cao, Xin-Min Li, Andrew J. Greenshaw, Yanbo Zhang.

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
