## [Decision Letter · Decision Letter 0]

5 Dec 2024

PONE-D-24-35477Suffering in Silence: Accessing Mental Health Care and Repetitive Transcranial Magnetic Stimulation (rTMS) for Peripartum Depression - A Qualitative StudyPLOS ONE

Dear Dr. Zhang,

Thank you for submitting your manuscript to PLOS ONE. After careful consideration, we feel that it has merit but does not fully meet PLOS ONE’s publication criteria as it currently stands. Therefore, we invite you to submit a revised version of the manuscript that addresses the points raised during the review process.

Please submit your revised manuscript by Jan 19 2025 11:59PM If you will need more time than this to complete your revisions, please reply to this message or contact the journal office at plosone@plos.org . Please include the following items when submitting your revised manuscript:

We look forward to receiving your revised manuscript.

Kind regards,

Mu-Hong Chen, M.D., Ph.D.

Academic Editor

PLOS ONE

Journal Requirements:

2. In the ethics statement in the Methods, you have specified that verbal consent was obtained. Please provide additional details regarding how this consent was documented and witnessed, and state whether this was approved by the IRB

“This study was supported by the University of Alberta Start-up Fund (RES0052505) awarded to Dr. Yanbo Zhang.”

4. In this instance it seems there may be acceptable restrictions in place that prevent the public sharing of your minimal data. However, in line with our goal of ensuring long-term data availability to all interested researchers, PLOS’ Data Policy states that authors cannot be the sole named individuals responsible for ensuring data access (http://journals.plos.org/plosone/s/data-availability#loc-acceptable-data-sharing-methods).

Data requests to a non-author instituti onal point of contact, such as a data access or ethics committee, helps guarantee long term stability and availability of data. Providing interested researchers with a durable point of contact ensures data will be accessible even if an author changes email addresses, institutions, or becomes unavailable to answer requests.

Reviewers' comments:

Reviewer's Responses to Questions

**Comments to the Author**

1. Is the manuscript technically sound, and do the data support the conclusions?

Reviewer #1: Partly

Reviewer #2: Yes

Reviewer #3: Yes

2. Has the statistical analysis been performed appropriately and rigorously?

Reviewer #1: N/A

Reviewer #2: N/A

Reviewer #3: Yes

3. Have the authors made all data underlying the findings in their manuscript fully available?

Reviewer #1: Yes

Reviewer #2: Yes

Reviewer #3: Yes

4. Is the manuscript presented in an intelligible fashion and written in standard English?

Reviewer #1: Yes

Reviewer #2: Yes

Reviewer #3: Yes

5. Review Comments to the Author

Reviewer #1: This qualitative study explores barriers and facilitators in accessing mental health care and specifically the use of repetitive transcranial magnetic stimulation (rTMS) for peripartum depression (PPD). Through interviews with individuals experiencing PPD and health providers, it identifies systemic and personal challenges and assesses attitudes towards rTMS as an emerging treatment. I have some comments as follows.

Introduction

1. “only a proportion of mothers develop PPD”. Please clearly state what percentage (proportion) this is.

Method

1. Please elaborate on the theoretical basis behind the interview design and the selection of specific questions, which can add rigor to the research method.

2. Please provide additional background information on the sampling strategy and selection criteria to clarify why these participants are representative of the broader PPD population. Otherwise, discussing the relevance and potential limitations of the sample can increase the transparency of the study.

Result

1. Overall, I feel that the paragraphs are too lengthy and contain too many quotations (narrative parts in italics). These quotations could perhaps be moved to supplementary materials or be condensed, such as by using tables or figures for presentation. This could aid readers in grasping the data more readily.

Discussion

1. Sample limited to Canada, potentially restricting generalizability to other healthcare contexts.

2. Self-selection bias may influence results, as participants interested in mental health might be more willing to share experiences.

3. Lack of longitudinal follow-up on participants’ subsequent treatment decisions, limiting understanding of rTMS's long-term acceptance or impact.

Reviewer #2: This manuscript points out the key issue in mental health care access for patients suffering from peripartum depression, and analyzes the topic from a qualitative standpoint through the use of interviews with potential patients. It also points out rTMS as an attractive and potential alternative to medical treatment for PPD. Although the study focuses on participants in Canada, this is an issue experienced in many countries globally and warrants increased attention. The research methodology is adequate, although additional clarifications should be made. The overall structure of the manuscript is lengthy, and condensation of some unnecessary text could help with reader understanding.

Major Comments:

1. Many of the quotes provided contain irrelevant information which interrupts the flow of the manuscript. Please condense each quote such that the identified theme is portrayed without additional distractions.

2. How many interviewers were there? Please provide credentials for the interviewer(s) (e.g. PhD, MD, etc.) in the manuscript.

3. Lines 124-130. How many researchers were involved in analyzing the interview transcripts? If more than one, were there any discussions that took place during the analysis, and how were disputes (if any) resolved?

4. Line 300. I fail to see the relevance between “importance of education” and the series of quotes on intrusive thoughts. Did the interviewer(s) confirm with the participants their failure to recognize these symptoms?

5. Did any of the participants receive treatment for depressive symptoms? Please provide the numbers for the people who did receive some kind of treatment, if any. If not, this would be a significant selection bias worth mentioning.

Minor Comments:

1. Line 82. Please fix the in-text reference formatting here.

2. Lines 135-137. Please clarify this sentence as it is hard to understand based on the current structure.

3. Line 300. Readers may be misled by the word “education” here to mean patient education level. Perhaps using “health education” or other specifications can clarify this heading.

4. You mention in the limitations the possibility for selection bias. Were any incentives associated with signing up for interview?

5. Perhaps providing identification to each quote may help with the overall presentation (e.g. participant 1, participant 2, etc.)

6. Line 569. What is SES?

Reviewer #3: This study identifies key barriers to mental care (and rTMS) for PPD. Understanding these barriers will support the development of an individualized care approach for the patients. I have some comments for consideration.

1. Introduction: I suggest add few potential mechanisms to explain the efficacy of rTMS in PPD.

2. “Existing research on rTMS as a treatment for PPD demonstrates the promise of rTMS as a safe and effective treatment for depression with onset during pregnancy [34-41] and postpartum [42-46], though there is still a need for more extensive randomized controlled trials to substantiate these findings further. There are several studies demonstrating the effect of rTMS in PPD but no RCTs in the cited references?

3. Please mention that in some countries, pregnancy is still contraindicated to receiving rTMS.

4. Method: did participant receive money in the present study? Please clarify it.

5. The study enrolled current and previous patients with PPD. Are there differences in barriers between “current” and “previous” patients with PPD?

6. Results: in table 1, please add demographic data of 36 participants including age, history of psychiatric disorder, etc.

7. Discussion: the objective of the study is to understand the barrier and improve mental care in PPD. I suggest adding one paragraph of “policy recommendation” based on your results.

6. PLOS authors have the option to publish the peer review history of their article (what does this mean? ). If published, this will include your full peer review and any attached files.

**Do you want your identity to be public for this peer review?** For information about this choice, including consent withdrawal, please see our Privacy Policy .

Reviewer #1: No

Reviewer #2: No

Reviewer #3: No

---

## [Author Response · Author response to Decision Letter 1]

19 Feb 2025

REVIEWER 1 COMMENTS:

This qualitative study explores barriers and facilitators in accessing mental health care and specifically the use of repetitive transcranial magnetic stimulation (rTMS) for peripartum depression (PPD). Through interviews with individuals experiencing PPD and health providers, it identifies systemic and personal challenges and assesses attitudes towards rTMS as an emerging treatment. I have some comments as follows.

Introduction

1. “only a proportion of mothers develop PPD”. Please clearly state what percentage (proportion) this is.

Response: Thank you for your comment. We added the appropriate percentages to the Introduction, please see page 3.

a. “While up to 85% of new mothers experience the transient and self-limiting condition “baby blues,” 29% of pregnant people, and 26% of people postpartum only a proportion of mothers develop PPD, making it one of the most common medical complications associated with this period.” (page 3)

Method

1. Please elaborate on the theoretical basis behind the interview design and the selection of specific questions, which can add rigor to the research method.

Response: We added additional details behind the interview design on page 6 under the Methods – Data collection heading.

a. “An iterative process was used for interview design to capture a comprehensive understanding of participants’ experiences with PPD, PPD treatment access, as well as perspectives and attitudes on rTMS. Initial iterations of the interview guide focused on rTMS only, but were insufficient to answer the research questions with adequate depth. In subsequent iterations, the flow of questioning was developed to naturally progress from participants’ experiences with PPD, to treatments they accessed, to a discussion about rTMS. Interview questions were designed to be open-ended, allowing participants to direct conversation toward what they deemed most significant. This flexibility aligns with principles of qualitative research that prioritize participant perspectives and emphasize their lived experiences.” (page 6)

2. Please provide additional background information on the sampling strategy and selection criteria to clarify why these participants are representative of the broader PPD population. Otherwise, discussing the relevance and potential limitations of the sample can increase the transparency of the study.

Response: We added additional information under the Methods – Participant recruitment heading to clarify selection criteria (page 6). We also added some further discussion under the “Limitations” section about limitations to the sampling strategy (Page 25-26).

Result

1. Overall, I feel that the paragraphs are too lengthy and contain too many quotations (narrative parts in italics). These quotations could perhaps be moved to supplementary materials or be condensed, such as by using tables or figures for presentation. This could aid readers in grasping the data more readily.

Response:Thank you for this important comment. We moved the majority of the quotations to Tables 2 and 3 to condense the results section and improve readability.

Discussion

1. Sample limited to Canada, potentially restricting generalizability to other healthcare contexts.

Response: Thank you, this is mentioned in the Discussion – Limitations section.

a. “Another limitation is the recruitment of only participants who live in Canada, possibly limiting the generalizability of the findings to countries with different healthcare systems and maternal mental health processes.” (Page 26)

2. Self-selection bias may influence results, as participants interested in mental health might be more willing to share experiences.

Response: This point has been expanded upon in the Discussion – Limitations section.

a. “Due to the voluntary nature of the sampling strategy, there is potential for self-selection bias, whereby participants interested in mental health or research participation, or those of a certain socioeconomic status or with lower caregiving burden, may be more likely to sign up for an interview.” (Page 25)

3. Lack of longitudinal follow-up on participants’ subsequent treatment decisions, limiting understanding of rTMS's long-term acceptance or impact.

Response: This is a helpful point, thank you. We have added a sentence in the Discussion – Limitations section to convey this:

a. “Further, there was no longitudinal follow-up on participants’ subsequent treatment decisions, limiting understanding of long-term acceptance and uptake of rTMS.” (Page 26)

REVIEWER 2 COMMENTS:

This manuscript points out the key issue in mental health care access for patients suffering from peripartum depression, and analyzes the topic from a qualitative standpoint through the use of interviews with potential patients. It also points out rTMS as an attractive and potential alternative to medical treatment for PPD. Although the study focuses on participants in Canada, this is an issue experienced in many countries globally and warrants increased attention. The research methodology is adequate, although additional clarifications should be made. The overall structure of the manuscript is lengthy, and condensation of some unnecessary text could help with reader understanding.

Major Comments:

1. Many of the quotes provided contain irrelevant information which interrupts the flow of the manuscript. Please condense each quote such that the identified theme is portrayed without additional distractions.

Response: Thank you for this important comment. We moved the majority of the quotations to Tables 2 and 3 to condense the results section and improve readability. The quotations that were kept within the text were shortened to only include relevant information.

2. How many interviewers were there? Please provide credentials for the interviewer(s) (e.g. PhD, MD, etc.) in the manuscript.

Response: There was a single interviewer, who holds a PhD. We have added this information to the manuscript in the Methods – Data collection section:

a. “The semi-structured interviews were conducted through Zoom by a single interviewer (PhD) until a point of data saturation was reached.” (Page 6-7)

3. Lines 124-130. How many researchers were involved in analyzing the interview transcripts? If more than one, were there any discussions that took place during the analysis, and how were disputes (if any) resolved?

Response: Two researchers were involved in thematic analysis; discussions took place throughout and disputes were resolved by reaching consensus. We have clarified the Methods – Data Analysis section, to more accurately reflect this.

a. “The interview transcripts were thematically analyzed using the Braun and Clarke six-phase framework [48]. Two researchers were involved in thematic analysis; discussions took place throughout and disputes were resolved by reaching consensus. In the first phase, data familiarization, the researchers read through and verified the interview transcripts. Subsequently, the anonymized transcripts were uploaded to the NVivo software, where transcripts were read more deeply and inductive codes were generated that captured significant and recurrent concepts and ideas within the data. The NVivo software organized all similarly coded quotes, facilitating the process of organizing various codes into themes. Themes were then reviewed and supplemented with relevant excerpts from the transcripts.” (page 7)

4. Line 300. I fail to see the relevance between “importance of education” and the series of quotes on intrusive thoughts. Did the interviewer(s) confirm with the participants their failure to recognize these symptoms?

Response: Thank you for this important point, The discussion on symptoms and intrusive thoughts was removed from the “importance of health education” section and moved to a section titled “Symptoms, intrusive thoughts, and suicidality”. Please see pages 9 to 10.

5. Did any of the participants receive treatment for depressive symptoms? Please provide the numbers for the people who did receive some kind of treatment, if any. If not, this would be a significant selection bias worth mentioning.

Response: Yes, many participants did receive treatment, although it was not easy to access for most. We have added the following paragraph to the Results – Demographic information:

a. Of the 32 participants who experienced depressive symptoms during the peripartum period, five did not receive any treatment. Three others did not receive treatment during their first pregnancy despite experiencing depressive symptoms but were able to access care in subsequent pregnancies, either by being more proactive or through the support of a nurse who facilitated the process. Among the 27 participants who did receive treatment (medication: 20 [74%]; psychotherapy: 16 [59%]), many described the significant challenges they faced in accessing care. These challenges will be explored further in the ‘Barriers and Facilitators in Navigating Peripartum Mental Health Care’ section. Additionally, treatment was often delayed until after delivery (n = 15; 56%), and for those already on medication prior to pregnancy, treatment was sometimes halted until postpartum.” (Page 8)

Minor Comments:

1. Line 82. Please fix the in-text reference formatting here.

Response: Thank you for noticing this error; a square end bracket has been added.

2. Lines 135-137. Please clarify this sentence as it is hard to understand based on the current structure.

Response: It has been revised to: “There were 36 interviewees in total; eight of these were health providers, four of whom also had lived experience of depressive symptoms during the peripartum period” for better clarity (Page 7-8).

3. Line 300. Readers may be misled by the word “education” here to mean patient education level. Perhaps using “health education” or other specifications can clarify this heading.

Response: The heading has been changed from “The importance of education” to “The importance of mental health education.”

4. You mention in the limitations the possibility for selection bias. Were any incentives associated with signing up for interview?

Response: There were no incentives associated with signing up for an interview. We have added a line in the Methods – Participant recruitment section (“There were no incentives associated with recruitment and participation”) to clarify this. The selection bias is attributable to other factors that we have now expanded upon in the Limitations section.

5. Perhaps providing identification to each quote may help with the overall presentation (e.g. participant 1, participant 2, etc.)

Response: We have added participant identification following all quotes.

6. Line 569. What is SES?

Response: “SES” has been revised to “socioeconomic status”

REVIEWER 3 COMMENTS:

This study identifies key barriers to mental care (and rTMS) for PPD. Understanding these barriers will support the development of an individualized care approach for the patients. I have some comments for consideration.

1. Introduction: I suggest add few potential mechanisms to explain the efficacy of rTMS in PPD.

Response: Thank you for your comment, we added the following to the introduction:

a. “There is evidence that rTMS exerts antidepressant effects by causing recurrent and consistent firing of coactive cortical neurons implicated in depression, thereby potentiating synaptic plasticity. rTMS may additionally have modulatory effects on meta-plasticity, the plasticity of synaptic plasticity” (Page 4)

2. “Existing research on rTMS as a treatment for PPD demonstrates the promise of rTMS as a safe and effective treatment for depression with onset during pregnancy [34-41] and postpartum [42-46], though there is still a need for more extensive randomized controlled trials to substantiate these findings further. There are several studies demonstrating the effect of rTMS in PPD but no RCTs in the cited references?

Response: Two of the cited articles are RCTS [37, 45], although they both have small sample sizes. More, and larger RCTs are certainly needed. The line has been revised to say:

a. “Existing research on rTMS as a treatment for PPD demonstrates the promise of rTMS as a safe and effective treatment for depression with onset during pregnancy [34-41] and postpartum [42-46]. However, there are only two randomized controlled trials, one conducted during pregnancy [37] and one postpartum [45], both with small sample sizes, underscoring the need for more extensive trials to validate these findings.” (Page 4)

3. Please mention that in some countries, pregnancy is still contraindicated to receiving rTMS.

Response: Thank you for this comment. While pregnancy and breastfeeding are often listed as exclusion criteria in neurostimulation-based research (including rTMS), they are not actual contraindications for rTMS treatment. In our search, we were unable to identify any specific countries that list pregnancy as a contraindication for receiving rTMS. If you are aware of any such countries, please let us know, and we would be happy to include this information.

4. Method: did participant receive money in the present study? Please clarify it.

Response: Participants did not receive any incentives, including monetary rewards, in the study. We have added a line in the Methods – Participant recruitment section (“There were no incentives associated with recruitment and participation”; page 6) to clarify this.

5. The study enrolled current and previous patients with PPD. Are there differences in barriers between “current” and “previous” patients with PPD?

Response: Thank you for this important comment. We considered whether participants had a current or previous history of PPD during our analysis; however, no significant differences were observed.

6. Results: in table 1, please add demographic data of 36 participants including age, history of psychiatric disorder, etc.

Response: We have added age and treatment received to Table 1. However, we did not specifically ask participants about their history of psychiatric conditions. This information was mentioned by some participants during the interviews but was not systematically collected. As a result, we cannot include it in the table, as it would not fully capture all the participants’ experiences. In future studies, it would be valuable to collect this information systematically using a demographic questionnaire.

7. Discussion: the objective of the study is to understand the barrier and improve mental care in PPD. I suggest adding one paragraph of “policy recommendation” based on your results.

Response: Thank you for this comment. A section titled “Policy recommendations” was added near the end of the discussion section (page 24-25).

Again, we are thankful for the constructive and helpful feedback. We hope that we have adequately addressed the reviewer’s comments and that our revised manuscript is now suitable for publication. We look forward to hearing from you. Thank you for your consideration.

---

## [Decision Letter · Decision Letter 1]

12 Mar 2025

Suffering in Silence: Accessing Mental Health Care and Repetitive Transcranial Magnetic Stimulation (rTMS) for Peripartum Depression - A Qualitative Study

PONE-D-24-35477R1

Dear Dr. Yanbo Zhang,

We’re pleased to inform you that your manuscript has been judged scientifically suitable for publication and will be formally accepted for publication once it meets all outstanding technical requirements.

Kind regards,

Mu-Hong Chen, M.D., Ph.D.

Academic Editor

PLOS ONE

Additional Editor Comments (optional):

Reviewers' comments:

Reviewer's Responses to Questions

**Comments to the Author**

1. If the authors have adequately addressed your comments raised in a previous round of review and you feel that this manuscript is now acceptable for publication, you may indicate that here to bypass the “Comments to the Author” section, enter your conflict of interest statement in the “Confidential to Editor” section, and submit your "Accept" recommendation.

Reviewer #1: All comments have been addressed

Reviewer #2: All comments have been addressed

2. Is the manuscript technically sound, and do the data support the conclusions?

Reviewer #1: Yes

Reviewer #2: Yes

3. Has the statistical analysis been performed appropriately and rigorously?

Reviewer #1: N/A

Reviewer #2: N/A

4. Have the authors made all data underlying the findings in their manuscript fully available?

Reviewer #1: Yes

Reviewer #2: Yes

5. Is the manuscript presented in an intelligible fashion and written in standard English?

Reviewer #1: Yes

Reviewer #2: Yes

6. Review Comments to the Author

Reviewer #1: The authors have thoroughly addressed my comments, and I believe this paper is now suitable for publication.

Reviewer #2: The authors have adequately addressed my concerns about the previous draft. I have no further comments.

7. PLOS authors have the option to publish the peer review history of their article (what does this mean? ). If published, this will include your full peer review and any attached files.

**Do you want your identity to be public for this peer review?** For information about this choice, including consent withdrawal, please see our Privacy Policy .

Reviewer #1: No

Reviewer #2: **Yes: ** Yang-Chieh Brian Chen

---

## [Editor Report · Acceptance letter]

PONE-D-24-35477R1

PLOS ONE

Dear Dr. Zhang,

I'm pleased to inform you that your manuscript has been deemed suitable for publication in PLOS ONE. Congratulations! Your manuscript is now being handed over to our production team.

Kind regards,

on behalf of

Dr. Mu-Hong Chen

Academic Editor

PLOS ONE